# Urinary Exosomal Cystatin C and Lipopolysaccharide Binding Protein as Biomarkers for Antibody−Mediated Rejection after Kidney Transplantation

**DOI:** 10.3390/biomedicines10102346

**Published:** 2022-09-21

**Authors:** Mi Joung Kim, Seong Jun Lim, Youngmin Ko, Hye Eun Kwon, Joo Hee Jung, Hyunwook Kwon, Heounjeong Go, Yangsoon Park, Tae-Keun Kim, MinKyo Jung, Chan-Gi Pack, Young Hoon Kim, Kyunggon Kim, Sung Shin

**Affiliations:** 1Division of Kidney and Pancreas Transplantation, Department of Surgery, Asan Medical Center, University of Ulsan College of Medicine, Seoul 05505, Korea; 2Department of Pathology, Asan Medical Center, University of Ulsan College of Medicine, Seoul 05505, Korea; 3Department of Convergence Medicine, Asan Institute for Life Sciences, Asan Medical Center, University of Ulsan College of Medicine, Seoul 05505, Korea

**Keywords:** kidney transplant, biomarker, exosome, urine, antibody-mediated rejection, Cystatin C, lipopolysaccharide binding protein

## Abstract

We aimed to discover and validate urinary exosomal proteins as biomarkers for antibody−mediated rejection (ABMR) after kidney transplantation. Urine and for-cause biopsy samples from kidney transplant recipients were collected and categorized into the discovery cohort (*n* = 36) and a validation cohort (*n* = 65). Exosomes were isolated by stepwise ultra-centrifugation for proteomic analysis to discover biomarker candidates for ABMR (*n* = 12). Of 1820 exosomal proteins in the discovery cohort, four proteins were specifically associated with ABMR: cystatin C (CST3), serum paraoxonase/arylesterase 1, retinol-binding protein 4, and lipopolysaccharide−binding protein (LBP). In the validation cohort, the level of urinary exosomal LBP was significantly higher in the ABMR group (*n* = 25) compared with the T-cell-mediated rejection (TCMR) group and the no major abnormality (NOMOA) group. Urinary exosomal CST3 level was significantly higher in the ABMR group compared with the control and NOMOA groups. Immunohistochemical staining showed that LBP and CST3 in the glomerulus were more abundant in the ABMR group compared with other groups. The combined prediction probability of urinary exosomal LBP and CST3 was significantly correlated with summed LBP and CST3 intensity scores in the glomerulus and peritubular capillary as well as Banff g + ptc scores. Urinary exosomal CST3 and LBP could be potent biomarkers for ABMR after kidney transplantation.

## 1. Introduction

Kidney transplantation is a cornerstone treatment for end-stage renal diseases, regardless of their etiology. Advances in immunosuppressive regimen and our understanding of the biological processes in transplantation across immunologic barriers have improved the success of kidney transplantation in the past decades; however, injury to the graft remains a critical hurdle to long-term graft survival [1,2,3,4]. Early detection of graft pathology and early intervention and treatment are crucial to a long-lasting functional graft. Currently, the gold standard in diagnosing kidney allograft pathology is graft biopsy, which is impractical for routine and repetitive monitoring due to its invasiveness, risk of complications, and limitation in histological interpretation [5].

Antibody-mediated rejection (ABMR) is the leading cause of late allograft failure after solid organ transplantation and contributes to two-thirds of kidney transplant failure [6,7]. ABMR was originally recognized as the presence of donor-specific antibodies (DSAs) and DSA-induced microcirculation injuries such as glomerulitis and peritubular capillaritis [6]. Accordingly, recent reports showed that DSAs are important prognostic factors for kidney allograft survival [8,9]. However, because antibodies are the end products of the immune response to an alloantigen, laboratory tests for DSAs provide little information about the underlying immunological processes and the presence of immunological memory [10]. Thus, alternative diagnostic measures that allow minimally invasive yet highly specific monitoring of allograft status are needed.

Exosomes are 40–200 nm-sized vesicles containing protein, mRNA, and miRNA that may serve as biomarkers of structural injury and renal dysfunction [11,12,13,14]. Particularly, exosomes are associated with antigen presentation to T cells [15,16] and tolerance induction [16,17]. These properties of exosomes such as immune modulation and cell-to-cell communication indicate that exosomes could be an advantageous source for studying the mechanisms of and discovering useful biomarkers for variable kidney diseases, including the injury processes in kidney transplantation [18]. In this study, we aimed to discover ABMR-specific urinary exosomal proteins by using a discovery cohort consisting of 36 kidney transplant recipients with biopsy-proven allograft pathology and validated their diagnostic utility in tissue, urinary serum, and urinary exosome in a separate cohort of 65 kidney transplant recipients.

## 2. Materials and Methods

### 2.1. Patients

All consenting kidney transplant recipients who had a post-transplant for-cause biopsy at Asan Medical Center between January 2017 and April 2020 for clinical reasons such as proteinuria or deterioration in function were recruited for this observational case–control study. Multiple solid organ transplant recipients and pediatric renal transplant recipients were excluded from this study. Urine samples were collected at 2–3 h before the indication biopsy. In addition, living kidney donors were enrolled as healthy non-transplant controls in this study, and urine specimens were collected just before donor nephrectomy.

Patients in the discovery cohort were classified into 5 groups according to their allograft histopathology: ABMR group (*n* = 12), T-cell mediated rejection (TCMR) group (*n* = 8), BK virus nephropathy (BKVN) group (*n* = 5), no major abnormality (NOMOA) group (*n* = 11), and donor group (*n* = 24). In BKVN, tubular cell injury and interstitial inflammation are often observed, potentially leading to the misdiagnosis of TCMR. Therefore, specimens with BKVN were included to find biomarker candidates to distinguish BKVN from TCMR. In the validation cohort, the ABMR, TCMR, NOMOA, and control groups had 25, 10, 19, and 11 recipients, respectively, of which the control group was composed of those who had stable renal function without a history of a rejection episode. This study received approval from the institutional review board of Asan Medical Center and all patients provided written informed consent. No organs/tissues were procured from prisoners. All living donor nephrectomies were performed at Asan Medical Center, while kidneys from deceased donors were procured at several centers in South Korea. All deceased donors were strictly managed by a government organization (KONOS, Korean Network for Organ Sharing) without any organ trade or illegal distribution.

### 2.2. Histopathology

For-cause biopsies were performed by an ultrasound-guided percutaneous puncture. Each biopsy was stained with hematoxylin and eosin, Masson trichrome, periodic acid-Schiff, and Jones-methenamine silver for interpretation. C4d immunohistochemistry (1:100, rabbit polyclonal; Cell Marque, Rocklin, CA, USA) was performed on paraffin-embedded, formalin-fixed specimens using the Ventana Medical Systems (Tucson, AZ, USA) according to the manufacturer’s protocol [19]. Every allograft biopsy specimen was evaluated for histologic characteristics according to the Banff 2015 criteria [20] by two renal pathologists (Y. Park and H. Go).

### 2.3. Urine Processing and Exosome Extraction

Exosomes were isolated by step-wise ultracentrifugation as previously reported [11,21] with a few modifications. Briefly, urine samples (≥30 mL) were collected by simple voiding and protease inhibitor (combination of 4-(2-aminoethyl) benzenesulfonyl fluoride hydrochloride (AEBSF-HCl, Sigma-Aldrich, St. Louis, MO, USA), leupeptin-hemisulfate (Sigma-Aldrich, St. Louis, MO, USA), and NaN3 (Sigma-Aldrich, St. Louis, MO, USA)) were immediately added. To remove urinary sediments, the samples were centrifuged at 4000 rpm for 15 min at 4 °C and the resulting supernatants were collected and stored at −80 °C until exosome extraction. Frozen urine samples (15 mL) were thawed and vortexed for 1 min, centrifuged at 17,000 *g* for 15 min at room temperature, and the resulting supernatant was collected and ultracentrifuged at 200,000 *g* for 70 min at room temperature using a Beckman Coulter Optima L-80xp ultracentrifuge, rotor SW40Ti (Beckman Coulter; Brea, CA, USA). The resulting supernatant was discarded and the pellet was dissolved in 11 mL DPBS for washing, which was ultracentrifuged again at 200,000× *g* for 70 min at room temperature. The supernatant was discarded and the resulting pellet of exosomes was used for protein isolation or quantification of exosome particles.

### 2.4. Sample Preparation for Proteome Analysis

Exosome fraction was lysed using lysis buffer (50 mM Tris-HCl (pH 8.5), 5% SDS) and quantified using BCA assay. A measure of 100 ug of exosome proteins was digested into peptides using the amicon-adapted enhanced FASP method [22], and salt was removed by the C18 desalting cartridge (Sep-Pak C18 1 cc, Waters, Milford, MA, USA). For details, refer to previously published papers [23,24]. Peptide mixture was dried and stored at −20 °C until proteome analysis. Prior LC-MS analysis, the dried peptide mixture was reconstituted with 0.1% formic acid.

### 2.5. Nano-LC-ESI-MS/MS Analysis

Exosome peptide mixture was separated using Dionex UltiMate 3000 RSLCnano system (Thermo Fisher Scientific, Waltham, MA, USA). The reconstituted peptide sample was injected into a C18 Pepmap trap column (20 mm × 100 μm i.d., 5 μm, 100 Å; Thermo Fisher Scientific) and separated by an Acclaim™ Pepmap 100 C18 column (500 mm × 75 μm i.d., 3 μm, 100 Å; Thermo Fisher Scientific) over 200 min (350 nL/min) using a 0–48% acetonitrile gradient in 0.1% formic acid and 5% dimethyl sulfoxide (DMSO) for 150 min at 50 °C. The LC was coupled to a Q Exactive mass spectrometer (Thermo Fisher Scientific, Waltham, MA, USA) with a nano-ESI source. Mass spectra were acquired in a data-dependent mode with 20 data-dependent MS/MS scans. The target value for the full scan MS spectra was 3,000,000 with a maximum injection time of 100 ms and a resolution of 70,000 at *m*/*z* 400. The ion target value for MS/MS was set to 1,000,000 with a maximum injection time of 50 ms and a resolution of 17,500 at *m*/*z* 400. Dynamic exclusion of repeated peptides was set to 20 s.

### 2.6. Database Searching and Label Free Quantitation

The MS/MS spectra from exosome peptide mixture were processed using the SequestHT on Proteome Discoverer (version 2.2, Thermo Fisher Scientific, Waltham, MA, USA) with the SwissProt human protein sequence database (May 2020). Precursor mass tolerance and MS/MS tolerance was ±10 ppm and 0.02 Da, respectively. A fixed modification was set as default including cysteine carbamidomethylation, and variable modifications were set to be n-terminal acetylation and methionine oxidation with 2 miscleavages. The false-discovery rates (FDR) were set at 1% for the peptides using “Percolator” [25]. Peptide filters which included the peptide confidence, peptide rank, score versus charge state and search engine rank were set at the default values. Label-free quantitation (LFQ) was performed using the peak intensity for unique and razor peptide of each protein and excluded peptides including methionine oxidation.

### 2.7. Biomarker Candidate Selection

The selection process of candidate biomarkers in the discovery cohort is summarized in Figure 1. To select specific biomarkers representative of each pathologic group, we selected proteins with significantly different mean scaled abundance compared with the NOMOA and DONOR groups using Student’s t-test. We also selected proteins that showed more than a 2.0-fold difference compared with the NOMOA group using the Volcano plot. Thereafter, the proteins that overlapped in the two analyses were selected. Among them, we selected up-regulated proteins with a higher abundance ratio (fold increase >2.5) and adjusted *p* value, and the results were visualized in a heatmap format using MeV software, version 4.9.0. We also selected overlapping proteins in the other two pathologic groups with a significant difference in abundance ratio (fold increase >2.5).

### 2.8. Evaluation of Exosomes

For immuno-gold labeling, we used a modified whole mount immuno-gold labeling method [26]. The specimens were viewed under a transmission electron microscope at 80 kV (Hitachi H-7600, Hitachi, Chiyoda City, Tokyo). The distribution of the size of exosomes was measured by dynamic light scattering using the Zeta Sizer Nano ZS apparatus (Malvern Instruments, Malvern, UK). The measurement runs were performed using standard settings. PE-conjugated anti-CD63 antibody (BD Biosciences San Jose, CA, USA) was used for exosome surface staining, and the stained samples were analyzed by flow cytometry (Beckman Coulter, Brea, CA, USA) and the FlowJo program.

### 2.9. Western Blot and Immunohistochemistry

Detailed methods for Western blot analysis (WB) and immunohistochemistry (IHC) are provided in the Appendix A. WB and IHC were performed using appropriate antibodies (Appendix A).

### 2.10. Statistical Analysis

Categorical variables are summarized as absolute and relative frequencies and the differences among values were analyzed using the Chi-squared test or Fisher’s exact test. Quantitative variables in each group are presented as the mean and standard error of the mean (SEM). Differences among means were compared using Student’s t-test, analysis of variance (ANOVA), or Kruskal–Wallis test. Bonferroni post hoc tests were performed when the differences between groups were statistically significant. To assess the ability of each biomarker for discriminating between NOMOA and ABMR, receiver operating characteristic (ROC) curves were generated and the area under the curve (AUC) was obtained for each biomarker. The predictive probability of combined biomarkers was assessed using binary logistic regression. Spearman correlation was used to calculate the correlation between two parameters and was reported with r values. Statistical significance was set at *p* value of 0.05. All statistical analyses were performed using IBM SPSS Statistics for Windows, version 21 (IBM Corp., Armonk, NY, USA), R software version 3.1.2 (R Foundation for Statistical Computing, Vienna, Austria), and GraphPad Prism 5 (GraphPad Software Inc. San Diego, CA, USA)

## 3. Results

### 3.1. Patient Characteristics

Baseline characteristics of patients in the discovery and validation cohorts are summarized in Table 1. The interval between kidney transplantation and for-cause biopsy was significantly longer in the ABMR group compared with other groups in the discovery cohort (*p* = 0.003) as well as in the validation cohort (*p* = 0.002). In addition, the proportion of recipients without an induction regimen was significantly higher in the ABMR group than in the other groups in both cohorts. The histologic characteristics of the groups were assessed according to the Banff classification (Appendix A).

### 3.2. Urine Exosome Characterization and Exosomal Proteins

The isolated exosomes were CD63-immunogold positive in electron microscopy and CD63-positive in flow cytometry (Figure 1A,B). The exosomes were approximately 100–200 nm in diameter (*n* = 12; 153.1 ± 22.94), which is compatible with the known size of exosomes (Figure 1C) [12,13,14]. The expression of exosomal markers (Alix and TSG101) was also identified by Western blot analysis (Figure 1D).

### 3.3. Discovery of Four Biomarkers for Discriminating Antibody-Mediated Rejection

The study design for identifying urinary exosomal proteins in the discovery and validation cohort is presented in Figure 2.

Exosomes in urine were derived from 5 groups and detected for label-free quantitative LC-MS/MS proteomic analysis. A total of 4193 urinary proteins were identified, of which 1820 exosomal proteins were identified in the five groups. We quantitatively compared the abundance of urinary exosomal proteins in each group with those in the NOMOA and DONOR groups. In total, 63, 108, and 53 exosomal proteins in TCMR, ABMR, and BKVN compared to NOMOA and DONOR groups, respectively, were significantly different from the others (Figure 3A). We selected 108 proteins with significantly different scaled abundance in the ABMR group compared with the NOMOA and DONOR groups and 834 proteins with more than a 2-fold difference compared with the NOMOA group (Figure 2B and Figure 3B). As a result, 46 proteins were shown to overlap between the two samples (Figure 3B). The fold change (FC) ratios of abundance in these 46 proteins were transformed by log_2_. The FC ratios were separated and visualized by Volcano plots according to log_2_ FC 1.5 (Figure 3C). Based on the abundance ratio, excluding proteins with missing values, we selected 18 significantly upregulated proteins with a log_2_ > 1.5 and adjusted *p* < 0.001 compared to the NOMOA group, and their abundance ratios were visualized by a heatmap (Figure 3D and Appendix A). Among the 18 up-regulated proteins, cystatin-C (CST3), serum paraoxonase/arylesterase 1 (PON1), retinol-binding protein 4 (RBP4), and lipopolysaccharide-binding protein (LBP) were shown to have significantly higher abundance ratios in the ABMR group compared with the TCMR and BKVN groups (Figure 3E). The ROC curves for each exosomal protein in differentiating ABMR from other allograft pathologies in the discovery cohort are shown in Figure 3F,G, and the AUC, 95% confidence interval (CI), cut-off value, specificity, sensitivity, positive predictive value (PPV), and negative predictive value (NPV) for each exosomal protein are described in Appendix A. LBP and CST3 had higher AUC values than PON1 and RBP4 did. LBP and CST3 were also present in the published common proteins of reported exosomal proteins in the Exocarta, Vesiclepedia, and Urinary exosomal protein database. Urinary exosomal LBP and CST3 were the most potent biomarker for differentiating ABMR from other pathologies.

### 3.4. Validation of the Four Biomarkers for Discriminating Antibody−Mediated Rejection

The abundance of the four urinary exosomal proteins in the validation cohort was quantified by Western blot. The fold change of LBP was significantly greater in the ABMR group than in all other groups (control; 3.24 ± 0.83, NOMOA;13.08 ± 8.23, and TCMR; 7.15 ± 5.08) (Figure 4A,C). The fold change of CST3 in the ABMR group (43.65 ± 13.19) was also significantly greater than in the control (9.08 ± 2.42) and NOMOA (9.87 ± 3.05) groups, but not significantly greater than the TCMR group (39.45 ± 31.18) (Figure 4A,C). The fold changes of RBP4 and PON1 were not significantly different among the groups (RBP; control, 7.59 ± 2.98, NOMOA, 18.83 ± 10.29, ABMR, 15.56 ± 6.41, PON1; control, 20.82 ± 15.33, NOMOA, 8.95 ± 7.16 ABMR, 33.64 ± 19.18) (Figure 4B,C). The ROC curves for each exosomal protein in differentiating ABMR from NOMOA in the validation cohort are shown in Figure 4D and Appendix A. When compared with CST3, LBP was a potent biomarker for differentiating ABMR from TCMR as well as from the control and NOMOA groups (Figure 4E).

### 3.5. Comparison of LBP and CST3 Expression in Kidney Allograft Tissue According to Pathologic Groups

Immunohistochemical stainings for LBP (Figure 5A) and CST3 (Figure 5C) were performed in 50 kidney allograft tissues of the validation cohort. The mean LBP score in the glomerulus in the ABMR group (1.84 ± 0.19) was significantly higher compared with those in the NOMOA group (0.11 ± 0.08) and the TCMR group (0.14 ± 0.14). In addition, the mean LBP scores in the peritubular capillary (0.56 ± 0.16), interstitium (1.16 ± 0.14), and tubules (0.32 ± 0.10) were significantly higher in the ABMR group compared with those in the NOMOA group (ptc;0.00 ± 0.00, i;0.39 ± 0.104, t;0.00 ± 0.00), but not significantly different from those in the TCMR (ptc;0.00 ± 0.00, i;0.39 ± 0.14, t;0.00 ± 0.00) group (Figure 5B). The mean CST3 score in the glomerulus in the ABMR group (0.80 ± 0.22) was significantly higher than those in the NOMOA (0.00 ± 0.00) and the TCMR groups (0.00 ± 0.00), whereas the CST3 scores in other lesions were not significantly different among the NOMOA, ABMR, and TCMR groups (Figure 5D) (NOMOA; ptc;0.00 ± 0.00, i;0.06 ± 0.06, t;0.00 ± 0.00 ABMR; ptc;0.28 ± 0.11, i;0.36 ± 0.13, t;0.08 ± 0.06, TCMR; ptc;0.00 ± 0.00, i;0.57 ± 0.37, t;0.00 ± 0.00).

To test the hypothesis that LBP and CST3 reflect ABMR-specific renal pathologies, we analyzed the correlations of protein expression scores in graft tissues with the histopathological findings according to the Banff classification (Figure 6). The Banff g score showed a strong correlation with the LBP score in the glomerulus (r = 0.917, *p* < 0.0001) and a moderate correlation with the CST3 score in the glomerulus (r = 0.555, *p* < 0.0001) (Figure 6A). Conversely, the summed LBP scores in the glomerulus and peritubular capillary were significantly higher when the Banff g + ptc score was ≥2 (*p* < 0.0001) (Figure 6B). Similarly, the summed CST3 scores in the glomerulus and peritubular capillary were significantly higher when Banff g + ptc score ≥2 (*p* = 0.0001) (Figure 6B). The summed LBP (r = 0.899, *p* < 0.0001) and CST3 (r = 0.598, *p* < 0.0001) scores in the glomerulus and peritubular capillary were also significantly correlated with Banff g + ptc scores (Figure 6C).

### 3.6. Levels of LBP, CST3, and RBP4 in the Urine and Serum

Protein levels of LBP and CST3 were examined by ELISA in matched whole urine and serum samples from WB-based validation (Appendix A). The concentrations of LBP, CST3, and RBP4 in whole urine without isolation of exosome and adjusted by the urine creatinine level were not significantly different among the groups. Urine; LBP; NOMOA, 0.02 ± 0.02, ABMR, 0.03 ± 0.01, TCMR, 0.07 ± 0.03, CST3; NOMOA 0.14 ± 0.05, ABMR, 0.93 ± 0.53, TCMR, 6.28 ± 3.99, RBP4; NOMOA, 6.8 ± 3.43, ABMR, 4.02 ± 2.41, TCMR, 5.92 ± 5.55 mg/g, NOMOA; 7214.22 ± 2484.38, ABMR; 11,720.43 ± 1220.25, TCMR; 15,543.10 ± 1865.58 ng/mL) (Appendix A). The levels of CST3 and RBP4 in serum were no differences between groups (CST3; NOMOA, 2435.12 ± 500.31, ABMR, 1890.68 ± 226.07, TCMR,1749.77 ± 196.81, RBP4; NOMOA, 50,802.00 ± 2291.80, ABMR, 58,761.05 ± 6912.12, TCMR,48,726.90 ± 7462.59). We also analyzed the association between urinary exosomal proteins and urinary proteins, and found that there was no association (LBP; r = 0.228, *p* = 0.38, CST3; r = 0.294, *p* = 0.252) (Appendix A).

### 3.7. Protein–Protein Interaction Analysis of LBP and CST3

The protein–protein interaction network for LBP and CST3 showed a total of 20 interacting genes. The common interactors between LBP and CST3 were shown to have functional relationships with each other, such as leukocyte activation, cell activation involved in immune response, vesicle-mediated transport, and myeloid leukocyte activation (Figure 7A). LBP and CST3 were responsible for immune response and cell activation by forming a network with the related factors such as CD14 (monocyte differentiation antigen), TLR4 (Toll-like receptor 4), and IL6 (interleukin-6). KEGG analysis for interaction partners showed that differentially expressed genes were abundant in pathways which are known to be associated with rejection such as Toll-like receptor [27], NF-kB [28], HIF-1 [29], apoptosis [30], and PI3K-AKT signals [31] (Figure 7B).

### 3.8. Combined Use of LBP and CST3 as Biomarkers for ABMR

Considering that LBP and CST3 have the highest AUC among four biomarkers, we combined these two biomarkers to improve the efficacy for discriminating ABMR from NOMOA. Estimating predicted probability from logistic regression was used to establish a diagnostic model that could evaluate whether the combination of the two biomarkers could improve the diagnostic value. The combination of two biomarkers has the highest AUC than each biomarker alone for discriminating ABMR from NOMOA (Figure 4D and Appendix A). However, it was not shown that combined biomarkers are superior to LBP alone for discrimination between ABMR and TCMR. The prediction probability of urinary exosomal LBP and CST3 was significantly correlated with summed LBP (r = 0.677, *p* = 0.003) and CST3 (r = 0.801, *p* = 0.0001) (Appendix A) scores in the glomerulus and peritubular capillary as well as Banff g + ptc scores (r = 0.712, *p* = 0.001) in the histologic lesion of ABMR in the validation cohort (Appendix A).

## 4. Discussion

This study shows that urinary exosomal proteins in kidney transplant recipients present a rich source of personalized biomarkers for monitoring kidney allografts. Particularly, we discovered and validated urinary exosomal proteins CST3 and LBP, which could be combined to serve as a potent composite biomarker of ABMR in kidney transplant recipients. Additionally, we found that urinary exosomal CST3 and LBP are strongly correlated with expression in the glomerulus and peritubular capillary of kidney allografts, especially in recipients with ABMR. To our knowledge, this study is one of the few that discovered and validated urinary exosomal biomarkers for ABMR in kidney allografts.

As in Fekih et al.’s study, biopsy specimens with borderline TCMR were excluded from this study because of the ongoing controversies regarding the clinical and immunological importance of borderline TCMR [14,32]. In addition, those with mixed borderline TCMR and ABMR were classified into the ABMR group considering the rarity of cases of pure ABMR without TCMR. Different from the study by Fekih et al., however, biopsy samples with mixed TCMR and ABMR were excluded from this study to discover ABMR-specific biomarkers.

In our study, LBP was strongly expressed in the glomerulus and peritubular capillary as well as tubules and interstitium in the kidney allograft specimens from patients with ABMR. In addition, urinary exosomal LBP was significantly increased in patients with ABMR. LBP is mainly produced in hepatocytes and plays a crucial role in the innate immune response [33]. The LPS-LBP complex binds to CD14 and the MD-4/MD-2 complex, thereby activating signal transduction pathways and releasing cytokines and pro-inflammatory molecules [34,35]. Single-nucleotide polymorphisms in the LBP gene are associated with infectious diseases, inflammatory disease, metabolic disorders, and malignancy [33]. However, only a few studies have reported the association between LBP and rejection after solid organ transplantation. Gerlach et al. showed that plasma LBP levels increased during acute and chronic rejections after intestinal transplantation in rats [36], and Freue et al. reported that seven plasma proteins including LBP were increased in recipients with biopsy-proven acute rejection [37]. A case–control discovery study by the Genome Canada Biomarkers in Transplantation Group showed that LBP was one of the proteomic signatures in the plasma during early acute kidney allograft rejection. Our study is thus meaningful in that it demonstrated the possible use of LBP as an ABMR-specific biomarker. CST3 was shown to be effective for estimating the glomerular filtration rate and is thus considered a potent biomarker for acute kidney injury [38,39]. Recently, it was suggested that urinary exosomal CST3 mRNA expression levels are representative of the changes in renal mRNA and protein expression and may thus be a more specific biomarker of renal damage than glomerular-filtered free CST3 [40]. Similar to LBP, our results showed that CST3 and its interaction with LBP may be useful as a combined biomarker for ABMR. Future studies are needed to decipher the biological mechanism by which CST3 and LBP are involved in acute rejection after kidney transplantation. This study is limited by its single-center design and the small number of samples, especially for TCMR. Due to rare BKVN cases during the study period, the BKVN group was not included in the validation cohort. Owing to the small number of ABMR samples, it is not feasible to verify a difference between acute ABMR and chronic active ABMR in terms of urinary exosomal biomarkers. Moreover, this was a cross-sectional study and did not include serial urine samples collected from various time points following kidney transplantation. Therefore, we are currently conducting a prospective study by collecting serial urine samples to assess how early these biomarkers can predict ABMR before pathologic diagnosis. For clinical application of exosomal urinary biomarkers, a modular microfluidic platform can be applied to isolate and enrich exosomes from urine samples of kidney transplant recipients. For quantification of specific exosomal proteins related to pathologic status, the exosomes on microbeads can be directly subjected to the molecular analysis [41].

## 5. Conclusions

In conclusion, we discovered and validated urinary exosomal proteins LBP and CST3 as potent non-invasive biomarkers for ABMR in kidney transplant recipients. Moreover, the 20 molecules and 5 complex molecular pathways that were significantly associated with the interactions between LBP and CST3 might play important roles in the biological progression of ABMR and become new treatment targets.

## Figures and Tables

**Figure 1 biomedicines-10-02346-f001:**
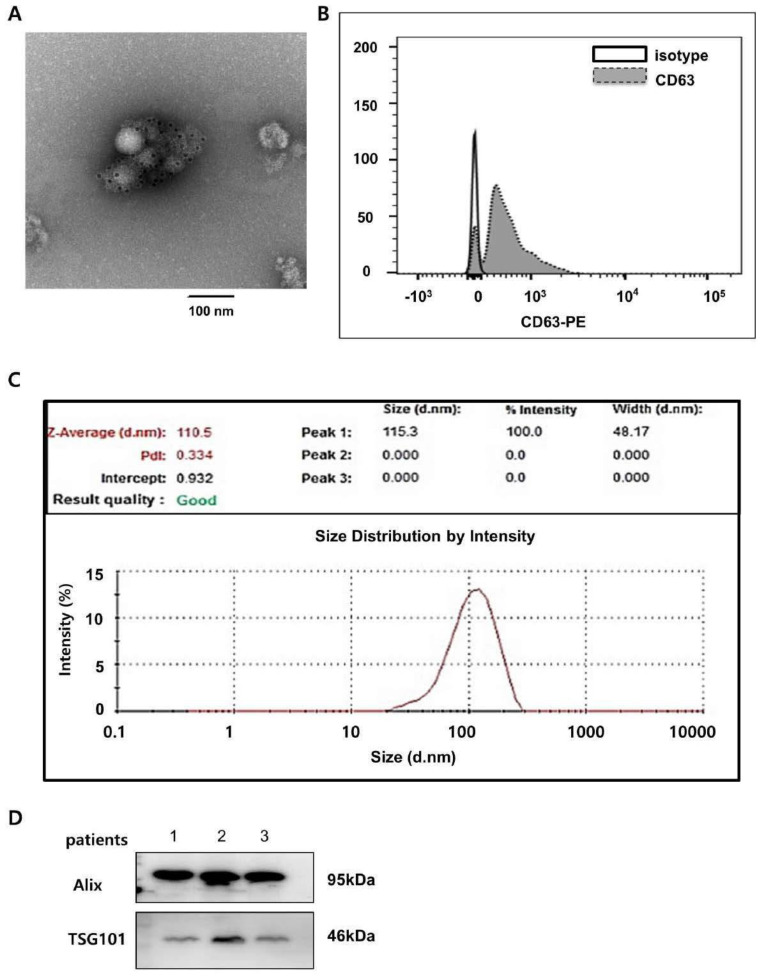
Urinary exosome characterization. (**A**) CD63 immunogold stain image of exosomes. Representative transmission electron micrographs of exosomes labeled with gold particles (black dots) are shown. (**B**) Flow cytometry analysis of the exosomal CD63 marker. (**C**) Size distribution of urinary exosomes analyzed by Zeta sizer. (**D**) Western blot analysis of the expression of exosomal markers Alix and TSG101 in urinary exosome proteins. ABMR, antibody−mediated rejection; TCMR, T cell−mediated rejection; NOMOA, no major abnormality; BKVN, BK virus nephropathy. FC, fold change.

**Figure 2 biomedicines-10-02346-f002:**
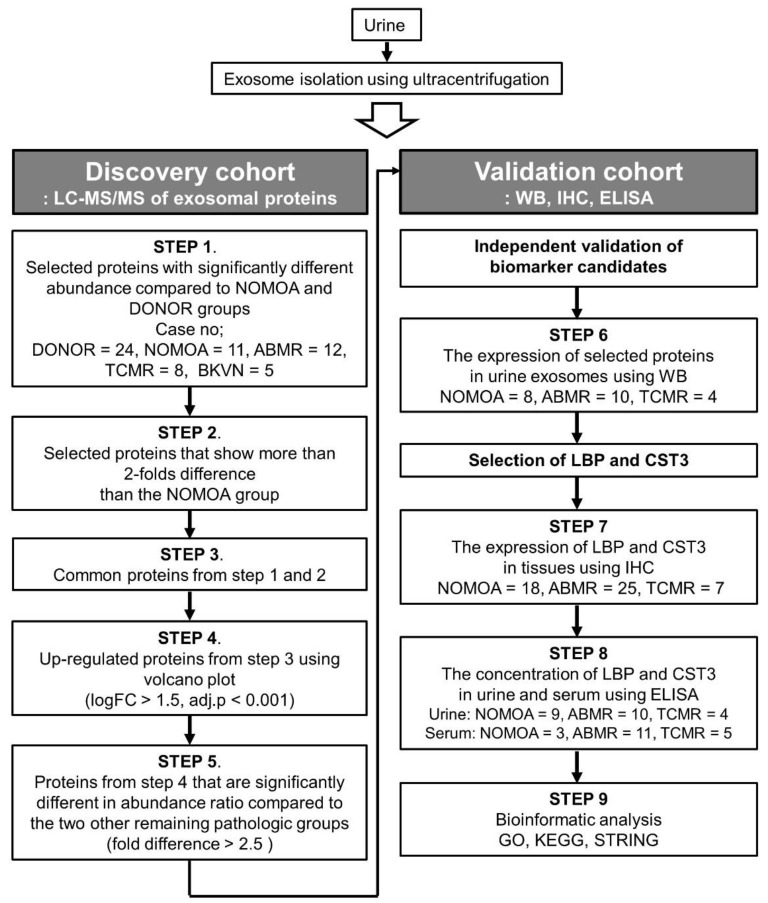
Study workflow for the discovery and validation of biomarkers representative of antibody-mediated rejection. LC-MS/MS, label-free liquid chromatography-mass spectrometry; ABMR, antibody-mediated rejection; TCMR, T-cell-mediated rejection; NOMOA, no major abnormality; BKVN, BK virus nephropathy; FC, fold change. WB, Western blot analysis; IHC, Immunohistochemistry; ELISA, Enzyme linked immunosorbent assay; LBP, lipopolysaccharide-binding protein; CST3, Cystatin-C; KEGG, GO, Gene Ontology; *Kyoto Encyclopedia of Genes and Genomes*.

**Figure 3 biomedicines-10-02346-f003:**
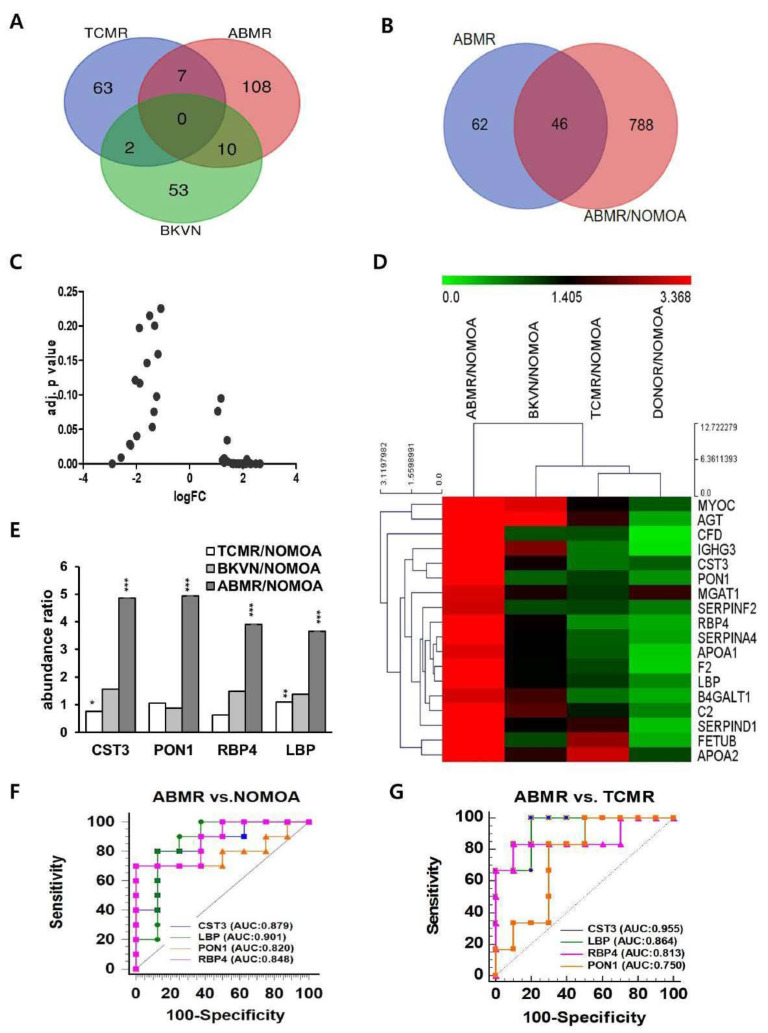
Discovery of proteomic biomarker candidates using proteomic analysis. (**A**) A total of 262 exosomal proteins were identified, of which 125 were identified in ABMR. (**B**) Forty−six proteins in the overlapping portion of ABMR and ABMR/NOMOA were selected as biomarker candidates for ABMR. (ABMR = 12, TCMR = 8, BKVN = 5). (**C**) A volcano plot was constructed using fold−change (LogFC) values and adjusted *p*-values for abundance ratio. The logFC > 0 represents up-regulated genes and logFC < 0 means downregulated candidates. (**D**) Heatmap showing the top 18 differentially expressed genes in upregulation after excluding missing values. The differentially expressed genes were selected on the basis of FC(log2) >1.5 and adj *p* < 0.001. (**E**) The relative expression levels of top 4 candidates in the ABMR and other pathologic groups for abundance ratio. * *p* < 0.05 ** *p* < 0.01 *** *p* < 0.001. (**F**,**G**) The area under the receiver operating characteristic curve was calculated to discriminate ABMR from other groups in the discovery cohort. ABMR, antibody−mediated rejection; NOMOA, no major abnormality. TCMR, T cell−mediated rejection; BKVN, BK virus nephropathy; AUC, area under the ROC curve.

**Figure 4 biomedicines-10-02346-f004:**
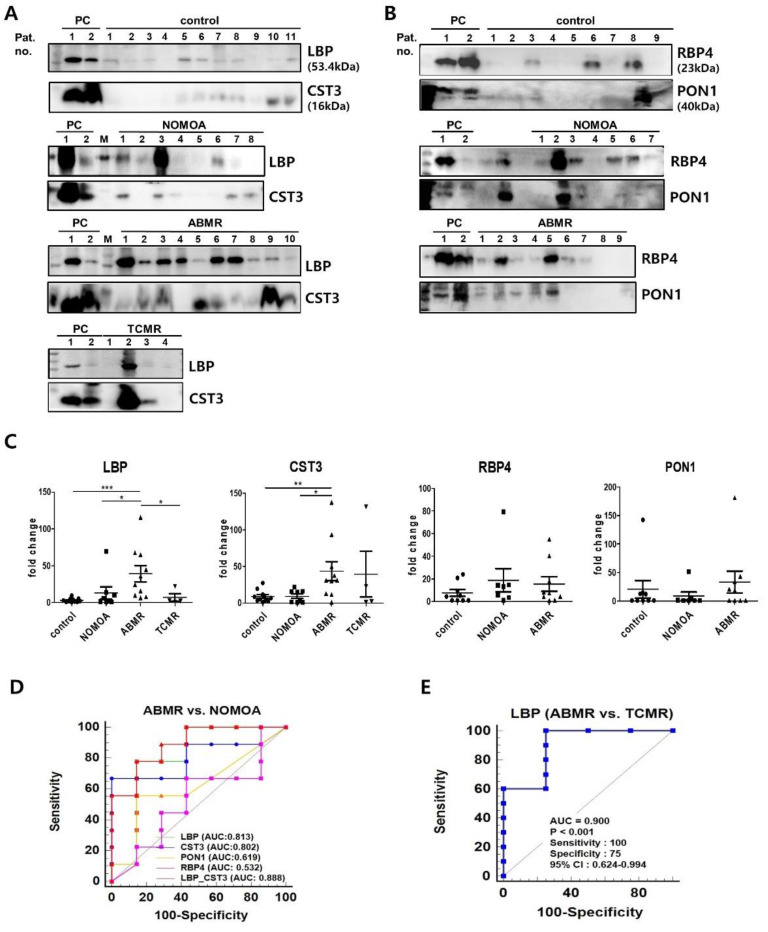
Validation of urinary exosomal biomarker candidates in urine exosome. Western blot analysis of LBP (**A**), CST3 (**A**), RBP4 (**B**), and PON1 (**B**) according to pathologic groups are shown. PC refers to positive control (pooling of 2 samples from an individual subject) groups (LBP and CST3; 11 control, 8 NOMOA, 10 ABMR, and 4 TCMR, RBP4 and PON1; 9 control, 7 NOMOA, and 9 ABMR). (**C**) Fold changes of signal intensities from the Western blots in (**A**,**B**). The results of Western blot analysis were assessed after normalization of means for the control group. Statistical significance was determined using two−tailed Mann−Whitney *U* test. * *p* < 0.05 ** *p* < 0.01 *** *p* < 0.001. (**D**,**E**) AUC of biomarker candidates in the independent validation cohort. D. ROC curves of single and two combined proteins between NOMOA and ABMR groups. (**E**) ROC curve of LBP for discriminating between ABMR and TCMR. ABMR, antibody−mediated rejection; NOMOA, no major abnormality. TCMR, T cell−mediated rejection; AUC, area under the ROC curve. CST3, Cystatin C; PON1, paraoxonase/arylesterase 1; RBP4, retinol−binding protein 4; LBP, lipopolysaccharide−binding protein; AUC, area under the ROC curve; 95% CI, 95% confidence interval.

**Figure 5 biomedicines-10-02346-f005:**
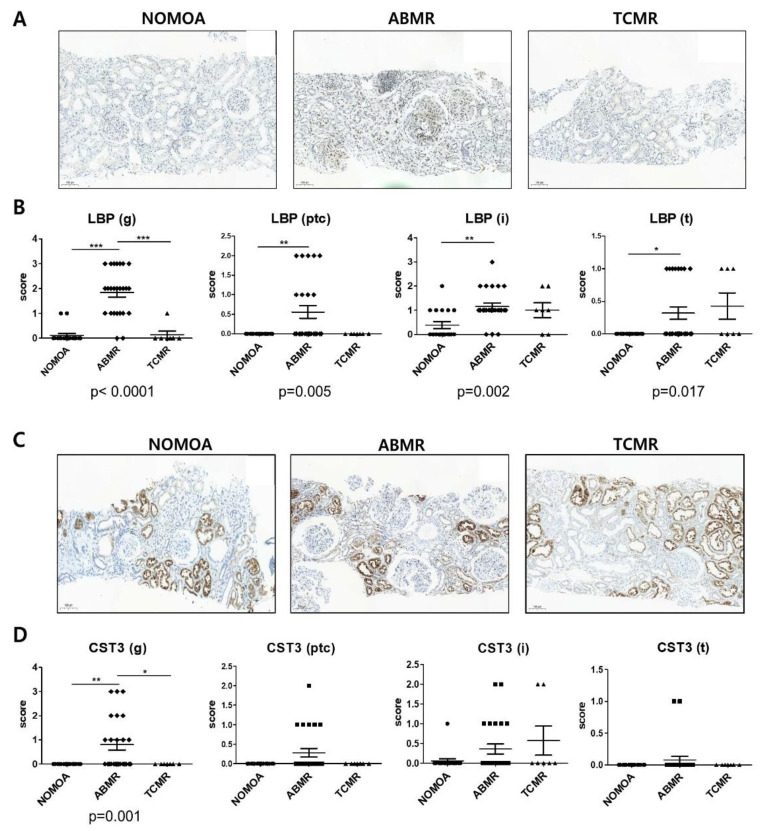
Validation for biomarker candidates in allograft tissues. (**A**) Immunohistochemistry (IHC) staining for LBP (visible as intense brown deposits) in representative biopsy specimens from the NOMOA (*n* = 18), ABMR (*n* = 25), and TCMR (*n* = 7) groups. (**B**) Expression of LBP corresponding to histopathologic scores in glomerulus (g), peritubular capillary (ptc), interstitium (i), and tubules (t) (×100 magnification). (**C**) IHC staining for CST3 (visible as intense brown deposits) in representative biopsy samples from the NOMOA, ABMR, and TCMR groups. (**D**) Expression of CST3 corresponding to histopathologic scores in g, ptc, i, and t (×100 magnification). Error bars indicate mean ± SEM. Kruskal–Wallis one-way ANOVA *p* values are indicated for each protein. Dunn’s Multiple Comparison test post hoc test significance values are indicated as * *p* < 0.05 ** *p* < 0.01 *** *p* < 0.001. 95% CI, 95% confidence interval. ABMR, antibody-mediated rejection; NOMOA, no major abnormality; TCMR, T-cell-mediated rejection; CST3, Cystatin-C; LBP, lipopolysaccharide-binding protein; g, glomeruli; ptc, peritubular capillaries; i, interstitium; t, tubules.3.6. Correlation between Banff lesion scores and biomarker scores of allograft.

**Figure 6 biomedicines-10-02346-f006:**
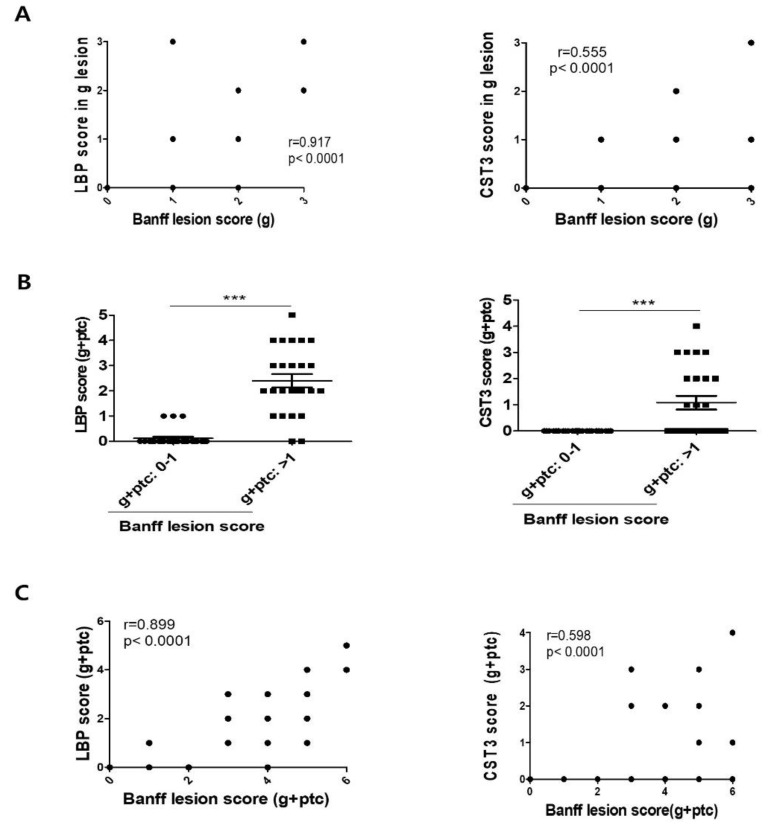
Association of urinary exosomal biomarkers with Banff scores in kidney allografts. (**A**) Correlation between the Banff lesion score (g) with LBP and CST3. (**B**) The statistical significance of microcirculatory inflammation (g + ptc score) was determined using two-tailed Mann–Whitney U test. *** *p* < 0.001. (**C**) Correlation between the Banff lesion score (g + ptc) with LBP and CST3. The results were categorized as strong (r = 0.7–1) and moderate (r = 0.5–0.7) according to the degree of association. Statistical significance was determined using two-tailed Mann–Whitney U test. *** *p* < 0.001. ABMR, antibody−mediated rejection; NOMOA, no major abnormality; TCMR, T cell−mediated rejection; CST3, Cystatin C; LBP, lipopolysaccharide−binding protein; g, glomeruli; ptc, peritubular capillaries.

**Figure 7 biomedicines-10-02346-f007:**
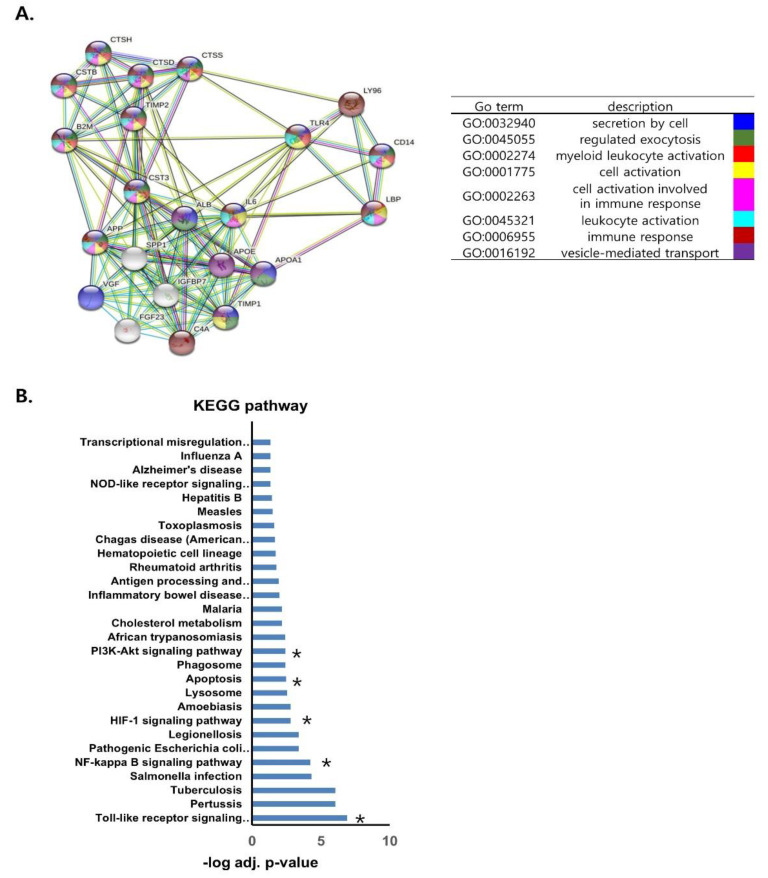
Protein−protein interaction network and co−expression analysis of LBP and CST3. (**A**) Protein−protein interaction network of the interactors using the STRING online database. The confidence score was set to 0.4 and the interactors were set to no more than 20 with first shell of interaction for human. Red, green, blue, purple, yellow, light blue, and black lines indicate the presence of fusion evidence, neighborhood evidence, co−occurrence evidence, experimental evidence, text mining evidence, database evidence, and co-expression evidence, respectively. (**B**) *Kyoto Encyclopedia of Genes and Genomes* (KEGG) pathway analysis of differentially expressed genes. A total of 28 pathways were significantly enriched. Asterisk indicates that the pathways are associated with allograft rejection according to previous studies. * pathways which are known to be associated with rejection.

**Table 1 biomedicines-10-02346-t001:** Baseline characteristics according to pathologic diagnosis.

	Discovery Cohort						Validation Cohort				
	ABMR	BKVN	TCMR	NOMOA	DONOR	*p*	ABMR	TCMR	NOMOA	Control	*p*
	(N = 12)	(N = 5)	(N = 8)	(N = 10)	(N = 24)		(N = 25)	(N = 10)	(N = 19)	(N = 25)	
Age (years)	52.2 ± 13.6	47.2 ± 8.6	53.8 ± 12.7	48.4 ± 10.2	38.5 ± 7.9	0.002	46.0 ± 10.8	51.0 ± 10.9	51.0 ± 12.6	50.6 ± 9.7	0.37
Female sex	5 (41.7)	1 (20.0)	1 (12.5)	4 (40.0)	13 (54.2)	0.326	7 (28.0)	3 (30.0)	4 (21.1)	13 (52.0)	0.011
BMI (kg/m^2^)	22.2 ± 2.6	23.7 ± 2.2	25.4 ± 4.3	21.6 ± 3.4	25.9 ± 3.8	0.011	22.0 ± 2.7	24.7 ± 2.7	22.7 ± 2.8	21.9 ± 3.0	0.351
Hypertension	1 (8.3)	0	2 (25.0)	0			2 (8.0)	0	1 (5.3)	4 (16.0)	
Diabetes	1 (8.3)	1 (20.0)	2 (25.0)	4 (40.0)			4 (16.0)	5 (50.0)	6 (31.6)	3 (12.0)	
Glomerulonephritis	1 (8.3)	0	0	1 (10.0)			5 (20.0)	0	3 (15.8)	6 (24.0)	
IgA nephropathy	0	1 (20.0)	1 (12.5)	1 (10.0)			3 (12.0)	2 (20.0)	1 (5.3)	2 (8.0)	
FSGS	0	0	0	1 (10.0)			0	0	1 (5.3)	0	
PCKD	0	0	1 (12.5)	0			0	0	2 (10.5)	1 (4.0)	
Unknown	6 (50.0)	3 (60.0)	1 (12.5)	3 (30.0)			9 (36.0)	2 (20.0)	5 (26.3)	8 (32.0)	
Others	3 (25.0)	1 (20.0)	0	2 (20.0)			2 (8.0)	1 (10.0)	0	1 (4.0)	
Serum creatinine (mg/dL)	3.25 ± 1.41	4.37 ± 2.80	1.61 ± 0.38	2.33 ± 1.12		0.005	2.01 ± 1.08	2.45 ± 1.31	1.979 ± 0.75		0.508
Albumin: creatinine ratio	1597 ± 1548	426.3 ± 643.3	290.7 ± 329.4	2816 ± 2170		0.059	1635 ± 1436	1422 ± 1737	732.6 ± 1027		0.236
Pre-emptive KT	2 (16.7)	1 (20.0)	0	2 (20.0)		0.594	2 (8.0)	1 (10.0)	1 (5.3)	7 (28.0)	0.05
Pre-transplant dialysis (month)	33.1 ± 40.2	30.0 ± 20.3	62.8 ± 71.6	48.4 ± 52.8		0.972	38.2 ± 54.0	49.4 ± 44.7	37.3 ± 44.3	36.2 ± 55.5	0.559
KT to biopsy (month)	182.8 ± 45.2	37.4 ± 62.1	64.3 ± 80.5	76.2 ± 67.9		0.003	105.3 ± 60.9	51.5 ± 59.1	68.9 ± 77.0	42.0 ± 36.5	0.002
Induction						0.025					0.046
None	10 (83.3)	0	2 (25.0)	3 (30.0)			6 (24.0)	2 (20.0)	3 (15.8)	1 (4.0)	
ATG	1 (8.3)	0	1 (12.5)	2 (20.0)			3 (12.0)	1 (10.0)	0	1 (4.0)	
Simulect	1 (8.3)	5 (100.0)	5 (62.5)	5 (50.0)			13 (52.0)	7 (70.0)	15 (78.9)	23 (92.0)	
Calcineurin inhibitor						0.318					0.112
Tacrolimus	2 (16.7)	5 (100.0)	2 (25.0)	5 (50.0)			17 (68.0)	4 (40.0)	13 (68.4)	19 (76.0)	
Cyclosporine	10 (83.3)	0	6 (75.0)	5 (50.0)			8 (32.0)	6 (60.0)	6 (31.6)	6 (24.0)	
Steroid maintenance	12 (100.0)	5 (100.0)	7 (87.5)	8 (80.0)		0.078	25 (100.0)	10 (100.0)	19 (100.0)	24 (96.0)	0.08
A/B/DR HLA mismatch	3.3 ± 1.5	3.2 ± 1.9	2.9 ± 2.0	3.1 ± 1.5		0.903	3.0 ± 1.5	3.7 ± 2.0	2.9 ± 1.6	3.0 ± 1.8	0.912
ABO-incompatible KT	1 (8.3)	1 (20.0)	0	0		0.241	3 (12.0)	3 (30.0)	5 (26.3)	0	0.009
DDKT	2 (16.7)	2 (40.0)	3 (37.5)	3 (30.0)		0.077	8 (32.0)	3 (30.0)	6 (31.6)	3 (12.0)	0.41
DSA detection	6 (50)	1 (20)	3 (37.5)	2 (25%)			15 (60)	4 (40)	4 (21.05)		
PRAI	18.67 ± 29.68	1.800 ± 2.68	11.00 ± 13.85	7.625 ± 8.93		0.557	30.32 ± 31.49	30.60 ± 36.98	7.84 ± 20.06		0.063
PRAII	43.42 ± 35.70	26.00 ± 35.80	41.63 ± 37.95	5.875 ± 15.82		0.144	65.32 ± 32.99	54.20 ± 39.77	25.16 ± 31.54		0.557

Values are mean ± standard deviation or n (%). ABMR, antibody−mediated rejection; BKVN, BK virus nephropathy; TCMR, T cell−mediated rejection; NOMOA, no major abnormality; BMI, body mass index; ESRD, end-stage renal disease; FSGS, focal segmental glomerulosclerosis; PCKD, polycystic kidney disease; KT, kidney transplantation; ATG, anti−thymocyte globulin; HLA, human leukocyte antigen; DDKT, deceased donor kidney transplantation; DSA, donor−specific antibodies; PRA, panel reactive antibody.

## Data Availability

The data presented in this study are available on request from the corresponding author. The data are not publicly available due to privacy of the participants.

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
