# Peer review of "Urinary Exosomal Cystatin C and Lipopolysaccharide Binding Protein as Biomarkers for Antibody−Mediated Rejection after Kidney Transplantation"

_biomedicines, 2022, doi:10.3390/biomedicines10102346_

Round 1
Reviewer 1 Report
Kim et al. aimed to discover and validate urinary exosomal proteins as biomarkers for antibody-mediated rejection (ABMR) after kidney transplantation. This is an interesting study presenting fairly novel results. However, the manuscript requires some adjustments before it is suitable for publication:
1. - the number of actual cases of ABMR analyzed should be included in the abstract
2. please provide rationale for including donors and cases of Polyoma BK nephropathy in the analysis
3. Table 1 – what does p value refer to? There are multiple groups and only a single p value provided? Which groups are you comparing?
4. - Figure 2 – according to the information provided in order to single out biomarkers representative of ABMR the Authors selected proteins with significantly different abundance compared to NOMOA and DONOR groups? What about TCR and BKV groups? Was there a significant difference?
5. - Why were there no cases of BKV in the validation cohort?
Reviewer 2 Report
I read with interest the article Urinary Exosomal Cystatin C and Lipopolysaccharide Binding Protein as Biomarkers for Antibody-Mediated Rejection After Kidney Transplantation by Mi et al.
The study evaluated the role of urinary exosomal proteins in kidney transplant recipients for monitoring kidney allograft. The study is well designed, showing the possible predictive role of urinary exosomal proteins to identify kidney rejection. The number of patients included is low, but still is the first step toward studies including more patients.
Would be interesting to have a section on clinical implications. In particular, would be interesting to know the limitation of the application of this method on a clinical daily bases (if there are any limitations) and the cost of the procedure.
Reviewer 3 Report
In this study, the authors aimed to discover antibody-mediated rejection (ABMR) -specific urinary exosomal proteins by using a discovery cohort consisting of 36 kidney transplant recipients with biopsy-proven allograft pathology and validated their diagnostic utility in tissue, urinary serum, and urinary exosome in a separate cohort of 65 kidney transplant recipients.
They concluded that, they discovered and validated urinary exosomal proteins lipopolysaccharide-binding protein (LBP) and cystatin-C (CST3) as potent non-invasive biomarkers for ABMR in kidney transplant recipients.
This article is original, and it is significant. But there is a minor problem.
Minor problem
The ABMR group contains acute/active ABMR and chronic active ABMR, but I think it is better to consider whether there is a difference between the two (acute/active ABMR and chronic active ABMR).
